# Primary Epstein–Barr Virus-Positive Mucocutaneous Ulcer of Esophagus: A Rare Case Report

**DOI:** 10.3390/jcm11164915

**Published:** 2022-08-21

**Authors:** Chunping Sun, Qingya Wang, Yujun Dong, Lin Nong, Yunlong Cai, Lihong Wang, Yuhua Sun, Wensheng Wang, Xinmin Liu

**Affiliations:** 1Department of Geriatric, Peking University First Hospital, Peking University, Beijing 100034, China; 2Department of Hematology, Peking University First Hospital, Peking University, Beijing 100034, China; 3Department of Pathology, Peking University First Hospital, Peking University, Beijing 100034, China; 4Department of Gastroenterology, Peking University First Hospital, Peking University, Beijing 100034, China

**Keywords:** EBV-positive mucocutaneous ulcer, esophagus, EBER, PD-L1, remission

## Abstract

Primary EBV-positive mucocutaneous ulcer (EBVMCU) is a rare and indolent disorder occurring in the oropharynx, skin, and gastrointestinal tract, with remission after removal of the immunosuppressive causes. We present a 69-year-old woman with heartburn, regurgitation of gastric acid, enlarged lymph nodes, and parotid glands. The endoscopic examination showed a circumscribed ulcer in the lower esophagus. A biopsy pathology indicated an esophageal EBV-associated lymphoproliferative disorder and a parotid gland/lymph node indolent B-cell lymphoma. Interestingly, the patient did not undergo any treatment, but the endoscopic ulcer improved significantly after more than 2 months. The last pathology showed EBV negativity, and EBVMCU was considered in combination with clinical and endoscopic manifestations. We followed up with the patient at 6 months, and the symptoms of acid reflux and heartburn had disappeared. Our case demonstrates that EBVMCU may occur in the esophagus with spontaneous regression.

## 1. Introduction

EBVMCU is an unusual kind of EBV-associated lymphoproliferative disease (EBV-LPDs). The disease manifests as shallow ulcers affecting the skin and mucosa. Only a few cases have been reported [1,2,3,4] in the esophagus. Due to its similarity or overlap with classical Hodgkin’s lymphoma and diffuse large B-cell lymphoma in morphology and immunophenotype, it is easily misdiagnosed [5]. EBVMCU has a self-limited course, with spontaneous regression after reduction or removal of immunosuppressive factors. Currently, patients with spontaneous remission of esophageal ulcers have not been reported. Here, we describe a patient with an esophageal ulcer who was finally diagnosed with EBVMCU after many pathological examinations. Without any treatment, the esophageal ulcer was spontaneously relieved. We reviewed this case to improve our understanding of the disease and to avoid misdiagnosis and overtreatment.

## 2. Case Presentation

A 69-year-old woman was admitted to the Department of Hematology, Peking University First Hospital, complaining of heartburn, regurgitation of gastric acid for 6 months in addition to night sweats, and a 6 kg weight loss. She did not present with dysphagia or odynophagia. She had no history of tuberculosis or neoplastic disease. The results of a physical examination on admission showed no palpable lymphadenopathy or organomegaly but left parotid gland enlargement. The patient was admitted to the Affiliated Hospital of Xinjiang Medical University before coming to our hospital. A gastroscopy showed crater-like ulcers in the esophagus and a biopsy pathology suggested a malignant tumor. Immunohistochemistry (IHC): AE1/AE3 (−), CD56 (−), CD3 (−), CD20 (−), ki-67 (90%). Positron emission tomography–computer tomography (PET-CT) showed bilateral multiple lymphadenopathies of the diaphragm, parotid gland, and esophagus involvement. The patient underwent a cervical lymph node dissection. Then, the paraffin block of the cervical lymph node was sent to the Hua Xi Kang Sheng Da Medical Laboratory for consultation, which suggested non-Hodgkin lymphoma and indolent lymphoma with pronounced plasma cell differentiation. IHC: CD20 (−), CD79a (+), CD10 (−), BCL-6 (−), BCL-2 (+), p53 (+), CD138 (+), MUM-1 (+), Kappa (+), Lambda (+), Ki-67 (5–20%). At the same time, the pathology of the esophageal biopsy suggested a lymphohematopoietic system tumor associated with an EBV infection. IHC: CD20 (−), CD79a (+), CD56 (−), Kappa (−), Lambda (+), CD3 (−), CD10 (−), Ki-67 (80%), in situ hybridization (ISH): EBV-encoded small RNA (EBER) (80%).

A gastroscopy from the outpatient department in our hospital on 2 September 2021 showed a 3.0 × 1.0 cm, shallow, longitudinal, and elliptical ulcer with an irregular border located in the posterior wall of the esophagus and 31 cm from the incisors (Figure 1A). The histopathology of the esophagus biopsy revealed inflammatory necrosis, exudation, granulation tissue, and large lymphoid cell proliferation scattered on the background of polytypic inflammatory cell infiltration in lamina propria. IHC: CD20 (−), CD79a (+), CD3 (−), CD56 (−), CD30 (−), CD138 (−), Kappa (+), Lambda (+), CD4 (−), CD8 (−), TIA-1 (−), Granzyme B (−), Ki-67 (60%). ISH: EBER++ (Figure 2A–E). IgH rearrangement was positive (Figure 3A), demonstrating a B-cell-derived lymphoproliferation disease. Programmed cell death-ligand 1 (PD-L1): 90 (number of positive tumor cells + number of positive immune cells)/total number of tumor cells × 100. Additionally, our hospital consulted the pathology of the esophageal biopsy and cervical lymph nodes from the previous local hospital, and the conclusions were consistent with indolent B-cell lymphoma.

The patient was admitted to the hematology ward of our hospital. The complete blood cell analysis showed a white blood cell count of 5.2 × 10^9^/L, a red blood cell count of 3.66 × 10^12^/L, a hemoglobin concentration of 102 g/L, and a platelet count of 276 × 10^9^/L. The serum lactate dehydrogenase (LDH) level was 149 U/L (normal range 100–240 U/L), the total serum protein level was 81 g/L (normal range 65–85 g/L), and the serum albumin level was 37.6 g/L (normal range 40–55 g/L). EBV-DNA in the plasma was negative. The EBV-DNA of lymphocytes in the blood was 907 copies/mL. Screenings for infectious diseases, including Human immunodeficiency virus (HIV), Cytomegalovirus (CMV), Herpes simplex virus (HSV), Varicella-zoster virus (VZV), and Human herpesvirus 8 (HHV-8), were all negative. Other laboratory findings were within the normal range. A bone marrow puncture showed no abnormal lymphocytes. ISH: EBER negative. There were no B cells with a clonal proliferation detected via bone marrow flow cytometry. 

Then, the patient underwent a parotid gland biopsy, and the pathology of the parotid gland showed a large amount of plasma-like cell hyperplasia infiltration. Most of the morphology was more mature, a few were naïve, and binuclear plasma cells could be seen with abundant small lymphocytes and histiocytes. IHC: CD3 (+), CD20 (+), CD79a (+), CD138 (+), Kappa (+), Lambda (+), MUM1 (+), CD10 (−), Bcl6 (−), IgG4 (−), Ki-67 (20%), ISH: EBER (−) (Figure 4), IgH rearrangement positive (Figure 3B). PD-L1: 90. As the monoclonal target regions of IgH gene rearrangement in pathological specimens of the esophagus and parotid were not consistent, we believed that the two lesions had different origins. Therefore, we suspected that the patient had both indolent B-cell lymphoma (lymph nodes, parotid gland involvement) and esophageal EBV-LPDs. The patient did not receive any treatment due to no indication of treatment. 

Two months later, the patient was admitted for a second gastroscopy with significantly relieved symptoms of acid reflux and heartburn. The ulcer of the esophagus was seen partly healed compared to before, and a scar could be seen around the ulcer (Figure 1B). The pathology of the esophageal biopsy showed inflammatory granulation tissue, necrotic inflammatory exudation, and patchy infiltration of medium-to-large lymphoid cells in the focal area. IHC: CD3 (−), CD20 (−), CD79a (+), CD56 (−), CD138 (−), MUM1 (+), p53 (+), C-myc (−), Ki-67 (20%). ISH: EBER (−) (Figure 2F–J). Combined with the medical history and previous multiple esophageal biopsies, EBVMCU was considered. The plasma EBV-DNA was still negative, and the lymphocyte EBV-DNA copy number in blood was 2990 copies/mL. Since the patient’s esophageal lesions showed a spontaneous remission, she did not receive any treatment during her hospitalization. We recommend that patients are followed up after discharge.

The patient has not received any treatment since her discharge from our hospital, and the symptoms of acid reflux and heartburn have disappeared, so the gastroscopy has not been reexamined.

## 3. Discussion

EBV is a common virus, also known as Human herpesvirus 4, which is transmitted mainly through saliva [6]. It constantly infects B cells in more than 95% of adults, leading to an asymptomatic lifetime carrier status [7]. When EBV disrupts the host immune balance, it can lead to a range of Epstein–Barr virus B-cell lymphoproliferative diseases (EBV-B-LPDs), ranging from indolent to lymphoma. Dojcinov et al. divided EBV-B-LPDs into five categories: infectious mononucleosis (IM); primary EBV+ diffuse large B-cell lymphoma (EBV+ DLBCL), not otherwise specified (NOS); EBVMCU; DLBCL-associated chronic inflammation (DLBCL-CI); and lymphomatoid granulomatosis (LyG) [8].

EBVMCU is a subtype of the newly increased EBV-LPDs with regression and an indolent course [5]. Ulcers mostly occur in elderly patients with a median age of 73 years (54–91 years), and these patients are predominantly female [1]. Most reported cases involve the oropharynx, followed by the skin and gastrointestinal tract (GIT). EBVMCU is relatively rare in the GIT, with approximately 30 cases at most reported in one article thus far [3,4,5,9,10,11,12,13,14,15,16,17,18,19,20,21,22,23,24,25,26]. Only approximately four cases have been reported to occur in the esophagus (Table 1) [27,28,29,30].

EBVMCU is primarily caused by an impaired immune function, such as age-related immune aging; iatrogenic immunosuppression, including methotrexate, cyclosporine, and azathioprine therapy; or steroid treatment. Additionally, it occurs in solid organ or bone marrow transplant recipients, in HIV-positive patients, after other lymphoma or tumor treatment, and infrequently in primary immunodeficiency [5,31,32]. In the immunocompromised hosts, local trauma or chronic mucosal irritation may favor the development of EBVMCU [5,32].

The disease presents as shallow, isolated, painful ulcers with clear borders, without associated systemic symptoms or involving any other site [33]. In the reported cases, an endoscope revealed various depths, linear and circular, and centimeter long or extensive ulcers in the esophagus [27,28,29,30]. Although EBER-ISH of biopsies have been strongly positive in reported cases, strangely, none of the patients had blood positive for EBV-DNA, and they were consistently negative during follow-up. Therefore, some scholars believe that the lack of quantifiable EBV viremia may be another feature of EBVMCU, but this should be validated in future cases [25]. 

Histologically, lesions present a polymorphic infiltration of lymphocytes and plasma cells, including large B-cell blasts that mimic Reed–Sternberg (RS) cells. At the same time, apoptotic monocytes, eosinophils, and some tissue cells can be seen scattered in it [5]. EBER is always strongly positive. These atypical EBV-positive cells generally show strong CD30 positivity. Variable B-cell surface antigens, such as CD20 and CD79a, are positive. PAX5, MUM1, and OCT2 are usually positive, whereas CD15 is positive in 50% of cases. CD10 and BCL6 are generally negative. Less than 50% of EBVMCU cases may harbor clonal IgH gene rearrangements or T cell receptor (TCR) gene rearrangement, which is supposed to be attributed to the clonal selection caused by EBV [5].

The patient in our case was characterized by acid reflux and heartburn as the primary clinical manifestations, and superficial esophageal ulcers were found via gastroscopy. The differential diagnoses contain infectious esophagitis caused by other pathogenic microorganisms such as CMV, VZV, or HSV, and Crohn’s disease. In this case, serological tests for the above viruses were negative, so infectious esophagitis can be ruled out. Crohn’s disease involving only the esophagus is rare [34]. The classical histological findings of Crohn’s disease are fissurating ulcers, acute and chronic inflammatory infiltrates, and sometimes noncaseating granulomas [34], which are not consistent with this case. Drug-induced ulcers can be excluded because the patient had no history of a specific medication. Multiple esophageal biopsy pathologies showed lymphoid cell hyperplasia with expression of CD79a and MUM1 but not CD20, CD3, or CD138. Moreover, IgH rearrangement was positive. Characteristically, EBER was strongly positive. Therefore, EBV-PLDs were considered several times in the esophageal pathology. Hodgkin’s lymphoma and EBV+ DLBCL, NOS, should also be excluded in this case. Hodgkin’s lymphoma is generally a tumor-forming lesion with transmural involvement in the digestive tract. The useful clue observed in EBVMCU is the high number of EBV-positive cells, which can be distinguished from Hodgkin’s lymphoma [5]. Meanwhile, EBV-positive DLBCL, NOS, and CD30 are usually positive. Unlike EBVMCU, EBV-positive DLBCL, NOS, displays a complete B-cell phenotype with the expression of CD20, CD19, PAX5, and CD79 alpha, and the transcription factors BOB1 and OCT2 are often positive [5,8,32]. Similar to EBVMCU, EBV-positive DLBCL, NOS, has an activated phenotype, being positive for MUM1/IRF4, and CD10 is negative [35]. In this case, the monoclonal positive target areas of IgH gene rearrangement in the pathological results of the esophagus and parotid glands from our hospital were different, so they were considered two different sources of disease. To distinguish these EBV-LPDs, it is crucial to understand the overall clinicopathological picture, particularly the extent of the disease. In our case, the diagnosis of EBVMCU was finally considered due to the esophageal lesions with spontaneous remission without any treatment when a reexamination gastroscopy was performed 2 months later. 

In our case, the patient’s pathology of an esophageal biopsy showed PD-L1 expression (SP263), which was similar to some of the literature reports such as Prieto et al. [16] and Daroontum et al. [36]. Specifically, Prieto et al. reported that PD-L1 was positive on both large cells and Hodgkin’s lymphoma Reed–Sternberg (HRS)-like cells in three patients with EBVMCU [16]. Intriguingly, Daroontum et al. described PD-L1 expression on tumor cells at initial onset, and multiple EBV-B-LPDs were generated after spontaneous regression of the disease in one EBVMCU case [36]. In contrast, Satou et al. found the deletion of PD-L1 expression on tumor cells in 7 cases of MTX-associated EBVMCU [37], which was consistent with the results of 13 cases of EBVMCU reported by Daroontum et al. [36]. Due to the small number of reported cases, the relationship between the expression of PD-L1 and EBVMCU, and whether it has a role in driving disease development into multiple B-LPDs, is yet to be concluded, and needs to be further explored in subsequent studies.

EBVMCU is characterized by a self-limited indolent course, generally with either a spontaneous remission or regression after reduction or discontinuation of immunosuppression within approximately 8 weeks [5,33]. The clinical outcome of our case is similar to that reported. In some reported cases, an antiviral treatment with acyclovir was initiated [28,29]. In other cases, only a symptomatic treatment was performed and regression was observed [27,30]. Spontaneous regression was observed in our case, which we speculate may be related to the relatively low EBV viral load, and the patient was not immunocompromised. However, there is still a lack of effective antiviral therapy and the role of acyclovir is unclear.

## 4. Conclusions

EBVMCU occurred in the esophagus of this case without any immunosuppressant or chemotherapy, and spontaneously achieved remission, which was rare in previous reports. The pathology of lymph nodes and a parotid gland biopsy were consistent in this patient, which was different from those with esophageal lesions. This poses a challenge to our differential diagnosis. What we can learn from this case is that EBVMCU has a variety of manifestations, such as esophageal ulcers, and may tend to spontaneously achieve remission. Moreover, the relationship between PD-L1 expression and EBVMCU in an esophageal biopsy, and whether it will lead to the development of B-LPDs, could be verified with follow-up visits. 

## Figures and Tables

**Figure 1 jcm-11-04915-f001:**
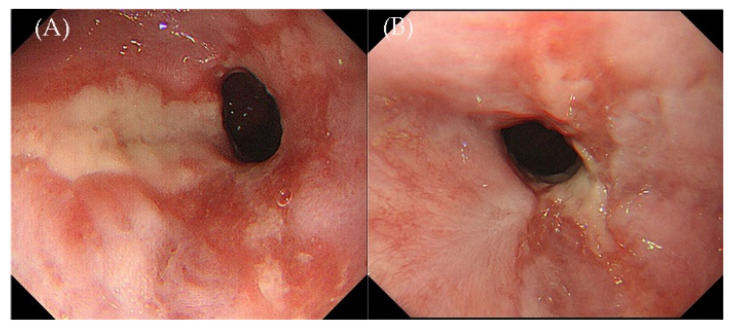
From left to right: (**A**) Gastroscopy from the outpatient department in our hospital on 2 September 2021. (**B**) Two months later, gastroscopy was performed in our hospital on 25 November 2021.

**Figure 2 jcm-11-04915-f002:**
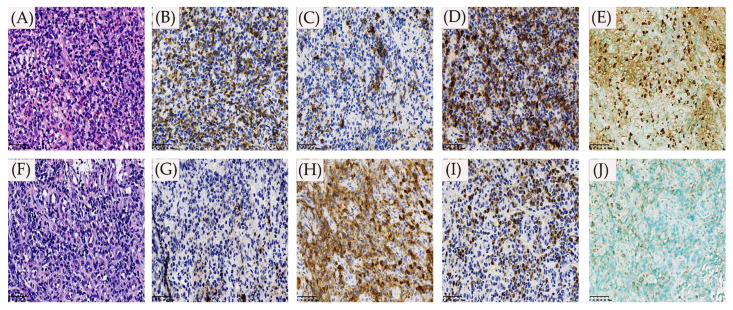
From left to right (**A**–**E**) Immunohistochemistry and in situ hybridization with Epstein-Barr virus-encoded small RNA (EBER) probe in the first esophageal biopsy in Peking University First Hospital. (**A**–**E**): HE × 40, CD3, CD20, CD79a, EBER. (**F**–**J**) Immunohistochemistry and in situ hybridization with Epstein–Barr virus-encoded small RNA (EBER) probe in the second esophageal biopsy in Peking University First Hospital. (**F**–**J**): HE × 40, CD20, CD79a, MUM1, EBER.

**Figure 3 jcm-11-04915-f003:**
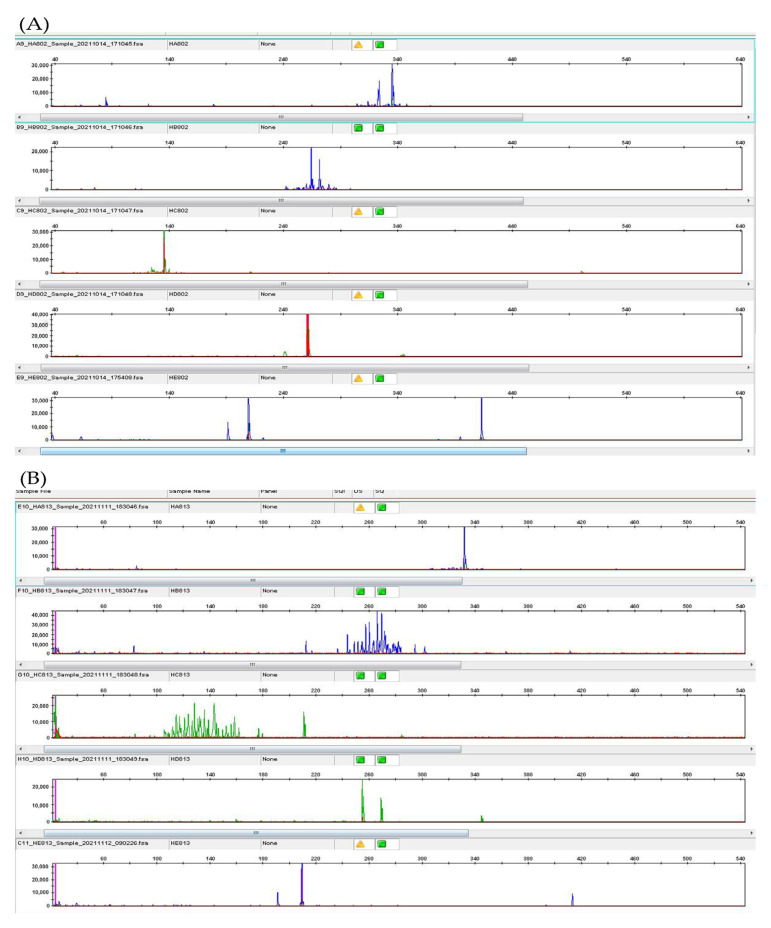
From top to bottom. (**A**) IgH rearrangement of esophagus biopsy: target segments FR1-JH, FR2-JH, FR3-JH, DH-JH display monoclonal. (**B**) IgH rearrangement of parotid biopsy: target segments FR1-JH, and DH-JH display monoclonal.

**Figure 4 jcm-11-04915-f004:**
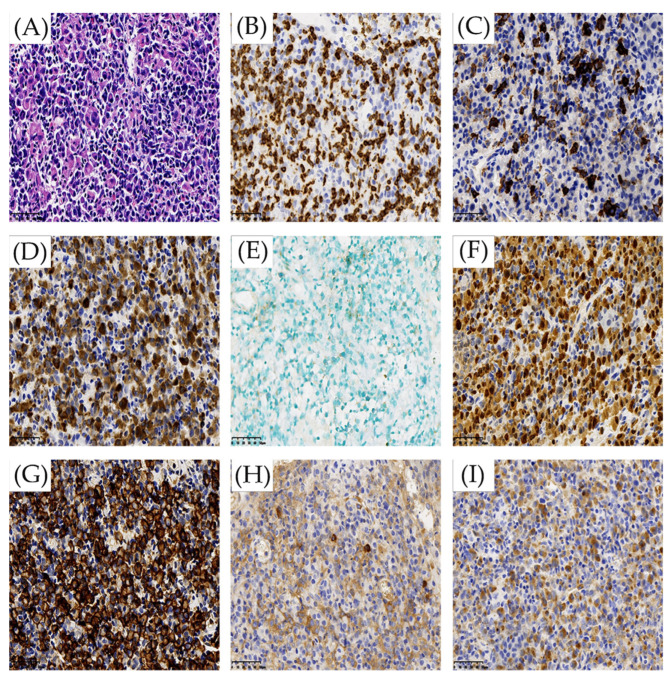
(**A**–**I**) Immunohistochemistry and in situ hybridization with Epstein-Barr virus-encoded small RNA (EBER) probe in the parotid biopsy in Peking University First Hospital. (**A**–**I**): HE × 40, CD3, CD20, CD79a, EBER, MUM1, CD138, Kappa, and Lambda.

**Table 1 jcm-11-04915-t001:** Summary of reported cases of EBVMCU in esophagus.

Age/Sex	Main Symptoms	Endoscopic Results	Diagnosis	Treatment
23/M [27]	Fever Diaphoresis Lethargy Weight loss Odynophagia Hematemesis	Multiple ulcerations	Positive Monospot and Paul Bunnell tests	Symptomatic
27/F [28]	Dysphagia Odynophagia	Multiple well-circumscribed Ulcerations differing depths	PCR on biopsy materials	Aciclovir for 5 days
48/M [29]	Fever Nausea Dysphagia	Denuded extensive ulcerations	PCR on biopsy samples	Aciclovir for 14 days
18/M [30]	Fever Muscle pain heartburn Epigastric pain dysphagia Odynophagia	Multiple, centimeter long, linear and circular ulcerations of various depth	EBV serology Positive EBV-specific ISH PCR analysis of biopsy specimens	Symptomatic

PCR: Polymerase Chain Reaction, EBV: Epstein–Barr virus, ISH: In situ hybridization.

## Data Availability

The datasets used during the current study are available from the corresponding author on reasonable request.

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
