# Peer review of "Primary Epstein–Barr Virus-Positive Mucocutaneous Ulcer of Esophagus: A Rare Case Report"

_jcm, 2022, doi:10.3390/jcm11164915_

Round 1
Reviewer 1 Report
EBV-related oesophagitis is an extremely rare type of oesophagitis and can be easily misdiagnosed. It rarely occurs due to reactivation in immunocompromised patients and has been observed as a manifestation of primary infection in only a few immunocompetent individuals. The disease has a good prognosis and can be treated symptomatically. EBV esophagitis should be considered in the differential diagnosis of infectious oesophagitis, Crohn's disease, and other causes of oesophageal ulcerations. Co-morbid causes of immunosuppression should also be ruled out. There are difficulties in diagnosis due to the lack of specific clinical and endoscopic symptoms and changes in the routine evaluation of biopsy. Therefore, in all cases, the final diagnosis is based on a combination of clinical, endoscopic and histopathological changes, and requires repeated examinations and additional examinations that are not performed in routine clinical practice. In addition, other diseases such as Crohn’s disease, CMV, ESV, HIV, Hodgkin’s lymphoma or diffuse B-cell lymphoma should be exluded. The presented case had no history of tuberculosis or neoplastic disease. Were other imaging and laboratory tests performed in the differential diagnosis to exclude other causes, including viral diseases such as CMV, HSV, VZV (e.g. serology IgM) or drug induced ulcers? Have you tested EBV serology (VCA/EBNA IgM and IgG) and EBV DNA on esophageal biopsies in addition to ISH? EBV esophagitis manifests with non-specific symptoms. Few previously reported cases (at a younger age) had dysphagia. Other reported symptoms include odynophagia, nausea, fever and weight loss. A 69-year-old woman presented in this study reported heartburn, acid regurgitation, and weight loss. Whether the patient reported swallowing problems? Moreover, the endoscopic results are not specific. In the previously described cases, numerous well-defined ulcers of various depths, linear and circular, centimeter long or extensive ulcers have been described. In some cases, antiviral treatment with acyclovir was initiated (Pape M, J Med. Case Rep 2009 and Annahazi A, Endoscopy 2011). However, effective antiviral therapy for EBV is lacking and any benefit from such treatment is unlikely. A spontaneous regression was observed in the study. In another case, only symptomatic treatment was used with good results (Tilbe KS, J Clin Gastroenterol 1986 and Lorentsen RD, BMJ Open Gastro 2021). Please consider adding the references mentioned above. An interesting observation of the authors is the PD-L1 expression. As they mentioned the relationship between the expression of PD-L1 and EBVMCU and its role in driving disease development into multiple B-LPDs remains to be concluded. Unfortunately, control gastroscopy was not performed after resolution of symptoms, and there were no follow-up visits. The authors reviewed this case to improve our understanding of the diesase and avoid misdiagnosis and overtreatment. Diagnosing EBV esophagitis is very challenging and requires complex testing and differential diagnosis. Considering only the few cases described in the literature so far, it would be ideal to present (in the form of a simple table) the general characteristics of the reported cases by gender, age, main symptoms, endoscopy results, diagnosis and treatment. All abbreviations used in the text should be explained.Author Response
Please see the attachment.

Reviewer 2 Report
The authors (Sun et al.) of the manuscript entitled “Primary Epstein-Barr Virus-Positive Mucocutaneous Ulcer of Esophagus: A Rare Case Report” investigated a case study on Epstein-Barr virus (EBV) positive mucocutaneous ulcer (EBVMCU) of oesophagus. EBVMCU is a recently documented B-cell lymphoproliferative disorder driven by latent EBV infection that results in distinct ulcerations in the oropharynx, gastrointestinal tract as well as skin. While under-reported, various case reports suggested that EBVMCU is mostly self-limited and is likely to resolve without any treatment. However, some EBVMCU cases can be incapacitating and persistent disorder, which may need aggressive therapy including surgical excision, immunotherapy, radiotherapy and chemotherapy to prevent further disease progression.
In this manuscript, the Authors presented a case of EBVMCU in a 69 year old woman suffering from heartburn and regurgitation of gastric acid for 6 months with enlarged left parotid gland. The authors observed the EBVMCU was resolved without any treatment.
Overall, the manuscript is well written with clear demonstration of figures. The work shall represent a good addition to the increasing knowledge of EBV association with different lymphoproliferative disorders and neoplasms.
I have no particular suggestion.
Minor Points:
In all the figures the subheading (A, B, C…) are missing.
